# The Expression of Adipogenic Marker Is Significantly Increased in Estrogen-Treated Lipedema Adipocytes Differentiated from Adipose Stem Cells In Vitro

**DOI:** 10.3390/biomedicines12051042

**Published:** 2024-05-09

**Authors:** Sara Al-Ghadban, Spencer U. Isern, Karen L. Herbst, Bruce A. Bunnell

**Affiliations:** 1Department of Microbiology, Immunology and Genetics, University of North Texas Health Science Center, Fort Worth, TX 76107, USA; spencer.isern@unthsc.edu; 2Total Lipedema Care, Tucson, AZ 85715, USA; kaherbst@gmail.com

**Keywords:** lipedema, hormone, estrogen, estrogen receptor (ER), ASCs, spheroids

## Abstract

Lipedema is a chronic, idiopathic, and painful disease characterized by an excess of adipose tissue in the extremities. The goal of this study is to characterize the gene expression of estrogen receptors (ERα and ERβ), G protein-coupled estrogen receptor (GPER), and ER-metabolizing enzymes: hydroxysteroid 17-beta dehydrogenase (HSD17B1, 7, B12), cytochrome P450 (CYP19A1), hormone-sensitive lipase (LIPE), enzyme steroid sulfatase (STS), and estrogen sulfotransferase (SULT1E1), which are markers in Body Mass Index (BMI) and age-matched non-lipedema (healthy) and lipedema ASCs and spheroids. Flow cytometry and cellular proliferation assays, RT-PCR, and Western Blot techniques were used to determine the expression of ERs and estrogen-metabolizing enzymes. In 2D monolayer culture, estrogen increased the proliferation and the expression of the mesenchymal marker, CD73, in hormone-depleted (HD) healthy ASCs compared to lipedema ASCs. The expression of ERβ was significantly increased in HD lipedema ASCs and spheroids compared to corresponding healthy cells. In contrast, ERα and GPER gene expression was significantly decreased in estrogen-treated lipedema spheroids. CYP19A1 and LIPE gene expressions were significantly increased in estrogen-treated healthy ASCs and spheroids, respectively, while estrogen upregulated the expression of PPAR-ϒ2 and ERα in estrogen-treated lipedema-differentiated adipocytes and spheroids. These results indicate that estrogen may play a role in adipose tissue dysregulation in lipedema.

## 1. Introduction

Lipedema is a hormone-related disease that affects approximately 11% of adult women in the United States. This disorder is characterized by the bilateral and symmetrical deposition of painful nodular subcutaneous fat in the lower extremities [1,2,3]. The bodily appearance of lipedema is often confused with obesity and lymphedema. Women with lipedema experience tenderness, heaviness of affected limbs, and difficulty in losing weight as the fat tissue is highly resistant to diet and exercise [4,5]. Lipedema patients can be categorized into four stages based on the severity of adipose tissue accumulation in the extremities. During the progression of the disease, patients may experience increasing levels of pain, swelling, and the accumulation of adipose tissue. In Stage 1, the skin appears smooth with small fat lobules. As the condition advances to Stage 2, indentations become noticeable with pearl-sized fat nodules beneath the skin’s surface. By Stage 3, the skin exhibits large extrusions and overhanging fat, leading to pronounced tissue deformities. Stage 4 represents a combination of lipedema and lymphedema, known as Lipo-lymphedema. It is important to note that lymphedema can develop concurrently with lipedema at any stage, but the presence of lymphedema alone does not qualify a case as lipedema [3,5]. Liposuction is the most effective treatment for decreasing fibrotic lipedema and fat, thereby improving mobility, which is essential for the quality of life of affected women [6,7,8]. While the etiology of lipedema remains unknown, hormones, genetic factors, inflammation, leaky dilated blood, and lymphatic vessels all contribute to the pathogenesis of lipedema [9,10,11,12,13,14].

Estrogen and estrogen receptors (ERs) control the distribution of female body fat and metabolism. ERα, ERβ, and G protein-coupled ER (GPER) proteins are differentially expressed in the upper and lower body fat in overweight-to-obese pre-menopausal women, indicating that estrogen influences subcutaneous adipose tissue (SAT) distribution through ER signaling [15,16]. ERα signaling regulates adipose-derived cells (ASCs) in their development and function [17]. Meanwhile, ERβ has direct anti-adipogenic effects on adipocytes, inhibiting the transcriptional activity of peroxisome proliferator-activated receptor gamma (PPARγ) [18]. ERα knockout (KO) mice have shown that a reduction in estrogen resulted in increased adipose tissue inflammation [19] with the upregulation of pro-inflammatory genes, namely interleukins (IL-1β, IL-6) and tumor necrosis factor-alpha (TNFα); the development of obesity; and insulin resistance. However, ERβ KO mice have shown no change in body weight or fat distribution compared to wild-type mice [18,20]. Estrogen also regulates leptin and lipoprotein lipase genes (LPL) [21] and increases angiogenesis and vascular endothelial growth factor (VEGF) expression [22], a phenomenon observed in the SAT of lipedema patients [9,10,23]. The G protein-coupled estrogen receptor, GPER, has been implicated in regulating body weight, metabolism, inflammation, and pain [16,24,25]. Studies using mouse models have demonstrated that mice lacking GPER exhibit increased adiposity (increased fat mass and adipocyte size) and reduced energy expenditure compared to wild-type mice. Furthermore, research indicates that the absence of GPER or estrogen receptor alpha (ERα) expression in mice leads to metabolic syndrome-like characteristics including inflammation, obesity, glucose intolerance, and insulin resistance [26,27,28]. The mechanisms by which estrogen acts through GPER are not yet fully understood. Investigating the interactions between estrogen receptors (ERs) and estrogen signaling pathways will contribute to a better understanding of their role in the development of lipedema.

Adipose-derived cells isolated from SAT have been widely used to study adipocyte differentiation and function in numerous estrogen-related diseases [29,30,31,32,33]. The biology of ASCs obtained from lipedema patients may also be regulated via estrogen and/or ERs, resulting in adipocyte hyperplasia and hypertrophy and increased inflammation and fibrosis in adipose tissues.

In this study, we defined the expression of ERs and ER-metabolizing enzymes: hydroxysteroid 17-beta dehydrogenase (HSD17B1, 7, B12), cytochrome P450 (CYP19A1), hormone-sensitive lipase (LIPE), enzyme steroid sulfatase (STS), and estrogen sulfotransferase (SULT1E1), which are markers in Body Mass Index (BMI) and age-matched non-lipedema (healthy) and lipedema ASCs.

The impact of estrogen treatment (17β-estradiol; E2) on the proliferation and adipogenic differentiation potential of ASCs at the transcription and translational levels in 2D monolayer cultures and spheroids was also determined. We hypothesize that hormones contribute to adipose tissue dysregulation in lipedema. Thus, studying the effect of estrogen on adipocyte differentiation will provide researchers insights into the mechanism involved in the development of the disease and will help direct future studies on hormonal therapy as a form of treatment for lipedema patients.

## 2. Materials and Methods

### 2.1. Cell Culture

Adipose-derived stem cells (ASCs) were cultured in Dulbecco’s Modified Eagles Medium (DMEM)/F12 (Hyclone, Logan, UT, USA), supplemented with 10% heat-inactivated fetal bovine serum (FBS, Hyclone, Logan, UT, USA) and 1% antibiotic/antimycotic (ThermoFisher Scientific, Waltham, MA, USA). The ASCs utilized in this study were isolated from lipoaspirates of subcutaneous adipose tissue from the thigh obtained from women undergoing elective liposuction. All subjects provided informed consent for inclusion before they participated in the study. The study was conducted per the Declaration of Helsinki and the protocols were approved by the Human Research and Protection Program at the University of Arizona (Institutional Review Board (IRB) protocol number: 1602399502) and Tulane University (IRB protocol number: 9189). ASCs isolated from healthy and lipedema patients were fully characterized individually before being pooled, as previously reported [34]. ASCs were used in passages 4–7 for the experiments. Table 1 summarizes the biological characteristics of ASCs used in this study.

### 2.2. Treatment of ASCs with 17β-Estradiol

To study the effect of 17β-estradiol (E2; Cat # E2758-1G, Lot #SLCJ5394, Sigma-Aldrich, St. Louis, MO, USA) on ASC proliferation and adipogenic differentiation, the cells were cultured in hormone-depleted (HD) medium [(DMEM/F12, no phenol red (Thermo Fisher Scientific, Cat #21-041-025, Waltham, MA, USA), charcoal-stripped (CS) FBS (ThermoFisher Scientific, Cat #12-676-029, Lot #2490726R) and 1% antibiotic/antimycotic] for 48 h and then treated with E2 at 100 nM for 72 h. In all the experiments, four groups were set as follows: ASCs cultured in DMEM/F12 without E2 as a control [Ctrl]; ASCs cultured in DMEM/F12 with E2 treatment [Ctrl + E2]; ASCs cultured in HD medium without E2 treatment [HD]; and ASCs cultured in HD medium with E2 treatment [HD + E2].

### 2.3. AlamarBlue Cell Proliferation Assay

ASCs were seeded at a density of 5 × 10^3^ cells/cm^2^ in a 96-well plate. Cell proliferation was assessed in culture on days 1, 7, 14, and 21. Cells were incubated with 20 μL of AlamarBlue reagent (Thermo Fisher Scientific, USA) for 2 h at 37 °C with 5% CO_2_. Fluorescence intensity was measured with excitation at 540 nm and emission at 600 nm on a Synergy™ HTX Multi-Mode Microplate Reader (BioTek, Waltham, MA, USA).

### 2.4. Flow Cytometry

For phenotypic analysis, cells were blocked with 1% BSA and 1% CD16/CD32 in 1x PBS and stained with the following antibodies at RT for 15 min: CD73 (Cat #: 550257, BD Biosciences, San Jose, CA, USA), CD90 (Cat #: 11-0909-42, Invitrogen, Waltham, MA, USA), and CD105 (Cat #: 17-1057-42, Invitrogen, Waltham, MA, USA). Cells were then fixed with 1% paraformaldehyde (PFA) and a total of 10,000 events were captured and analyzed with BD Accuri™ C6 Plus (BD Biosciences, USA).

### 2.5. Adipogenic Differentiation

ASCs were seeded at a density of 2 × 10^4^ cells/cm^2^ and grown in DMEM/F12. Before differentiation, the medium was replaced with hormone-depleted (HD) media or kept in DMEM/F12 medium for 48 h, and the cells were either treated or not with E2 for 72 h. After treatment, the medium was changed to adipogenic differentiation-inducing medium (AdipoQual, Obatala Biosciences, New Orleans, LA, USA) to differentiate the cells in the presence and absence of E2. Undifferentiated control cells were kept in the DMEM/F12 medium. The medium was changed every 3 days until day 21 (D21). For staining, control and differentiated cells were fixed with 4% PFA for 30 min and stained with filtered Oil Red O (Sigma, USA) for 15 min, followed by multiple rinses with 1x PBS. These cells were then visualized using EVOS™ M5000 Imaging System (Thermo Fisher Scientific) at 10× and 20× magnification. The absorbance of the Oil Red O eluted by adding 100% isopropanol was measured at a 584 nm wavelength using spectrophotometry. Differentiation values are reported as a percent of undifferentiated control cells. For transcriptional analysis, plates were washed twice with 1x PBS and stored at −80 °C for RNA extraction.

### 2.6. Formation of ASC Spheroids

ASCs were seeded at a density of 30 × 10^3^ cells/cm^2^ in 12-well plates coated with Ultrapure Agarose solution (1.5% *w*/*v*, Technologies, Carlsbad, CA, USA) dissolved in basal medium DMEM/F-12. The plates were placed on an orbital shaker at 50 rpm. ASCs aggregated in suspension (forming spheroids) and the assembly process was examined using EVOS™ M5000 Imaging System on days 2 and 10 with images at 10× magnification. The diameter of the spheroids was measured with ImageJ software (Version 1.54i) (National Institutes of Health, Bethesda, MD, USA, http://imagej.nih.gov/ij/, accessed on 3 March 2024).

### 2.7. Quantitative Polymerase Chain Reaction (qPCR)

Total RNA from ASCs was extracted using an RNA extraction kit (Qiagen, Germantown, MD, USA). One microgram of mRNA was used for cDNA synthesis with an Applied Bioscience purification kit (Thermo Fisher Scientific, USA). qRT-PCR was performed using the instructions from the SYBR Green qPCR SuperMix (Bio-Rad, Hercules, CA, USA) according to the manufacturer’s instructions. Oligonucleotide primers were designed using the vendor’s software (IDT, Coralville, IA, USA). Table 2 lists the primer sequences used for qRT-PCR. PCR conditions: 2 min at 95 °C and 40 cycles of 15 s at 95 °C and 30 s at 60 °C. The target and reference genes were amplified in separate wells. All reactions were performed in duplicate. The 2^(−∆∆CT)^ method was used to quantify gene expressions and normalized data to GAPDH, which was used as an internal control.

### 2.8. Western Blot Analyses

Capillary Western analyses were performed using the ProteinSimple Jess System (V 2.5). Cells were lysed with RIPA lysis buffer (Cat #: 89900; Thermo Fisher) supplemented with 1X protease inhibitor (Cat #: 1862209; Thermo Fisher). Protein samples were quantified using the bicinchoninic acid assay (BCA, Cat #: 23225; Thermo Fisher), and a total of 0.4 mg of protein lysate was loaded onto the plate along with the following primary antibodies for ERα (1:10; Cat #: AF5715, R&D systems, Minneapolis, MN, USA), ERβ (1:10; Cat #: NB200-305, Novus Biologicals, Bio-Techne, Minneapolis, MN, USA), PPARγ (1:100; Cat #: 2443; Cell Signaling Technologies, Danvers, MA, USA), and GAPDH (1:300; Cat #:2118; Cell Signaling Technologies, Danvers, MA, USA). After loading the plate according to the manufacturer’s instructions, the separation electrophoresis and immunodetection steps occur in the fully automated capillary system. Jess Western data were analyzed using Compass for Simple Western software (V 2.5) (ProteinSimple, Bio-Techne, Minneapolis, MN, USA). The area under curves from chemiluminescence chromatograms was used to determine the relative number of proteins. Expression levels of all proteins were normalized to GAPDH for the 12–230 kDa Separation Modules (ProteinSimple, Bio-Techne, Minneapolis, MN, USA). Pseudo-blots, generated via the compass software from the high dynamic range 4.0, are presented with each protein of interest.

### 2.9. Statistical Analysis

GraphPad PRISM 8 was used for all statistical analyses. The Mann–Whitney U test was used to determine the differences between the two ASC groups. One-way ANOVA and Tukey’s post hoc test were used to analyze the differences between the four groups. Asterisks (*) indicate statistical significance: * *p* < 0.05; ** *p* < 0.01; *** *p* < 0.001; **** *p* < 0.0001.

## 3. Results

### 3.1. Estrogen Increased the Proliferation and the Expression of Stemness Markers in Healthy ASCs in 2D Monolayer Culture

To study the effect of estrogen (E2) treatment on ASC proliferation and stemness, ASCs were seeded in HD media before treatment. The data show a significant decrease in the proliferation rate of both healthy and lipedema ASCs treated with HD media alone as compared to cells cultured in DMEM-F12 (control) media (Figure 1A). Interestingly, E2 treatment did not affect the proliferation rate of ASCs cultured in DMEM-F12 media (Figure 1B); however, it significantly increased the proliferation rate of healthy but not lipedema ASCs cultured in HD media, indicating that lipedema ASCs are not responsive to estrogen treatment (Figure 1C).

E2 treatment also increased the expression of mesenchymal stem cell (MSC) markers (CD73, CD90, and CD105) in healthy ASCs but not in lipedema ASCs (Figure 1D–F). Additionally, HD media significantly increased the expression of CD73 in healthy ASCs compared to lipedema ASCs and cells cultured in control media (Figure 1D).

### 3.2. Estrogen Increased the Expression of ERs in ASCs in 2D Monolayer Culture, but It Decreased the Expression of ERα and GPER in Lipedema Spheroid

E2 treatment significantly increased the gene expression of ERβ (~3-fold) in both HD-treated healthy and lipedema ASCs compared to untreated HD cells. In contrast, GPER gene expression decreased in HD-treated healthy ASCs compared to untreated HD cells (Figure 2A). It is worth noting that HD media alone significantly increased the expression of ERs in both healthy (ERα, 3-fold; ERβ, 2-fold) and lipedema (ERα, 3-fold; ERβ, 4-fold) ASCs compared to untreated control cells. ERα and ERβ expression is significantly higher at protein expression level in HD-treated lipedema ASCs (2-fold) than in untreated control healthy and lipedema ASCs (Figure 2B,C, Appendix A).

Interestingly, the expression of ERβ (2-fold) at both the gene and protein levels was significantly increased in HD lipedema ASCs and spheroids (Figure 3A,B, Appendix A) compared to untreated control lipedema and healthy cells. In contrast, ERα and GPER (0.5-fold) gene expression was significantly decreased in estrogen-treated lipedema spheroids compared to control cells and healthy spheroids (Figure 3A). The difference in the ER expression in E2-treated ASCs between 2D monolayer and spheroids is due to differences in culture conditions, the microenvironment, cellular interactions, and effects of the treatment.

### 3.3. Estrogen Significantly Increased the Expression of HSD17B7, LIPE, and STS in Lipedema ASCs and CYP19A1 in Healthy ASCs in 2D Monolayer Culture

E2 treatment significantly increased the gene expression of HSD17B7 in both HD-treated healthy (~1.7-fold) and lipedema (~2-fold) ASCs compared to untreated control cells (Figure 4B). Interestingly, HSD17B7 gene expression is higher in HD-treated lipedema cells than in healthy cells, suggesting a higher conversion of estrone (E1) to estradiol (E2) in lipedema.

E2 treatment significantly increased the gene expression of LIPE (3-fold) and STS (1.5-fold) in HD-treated lipedema cells compared to untreated cells (Figure 4C,D). E2 treatment also significantly increased the gene expression of CYP19A1 (~10-fold) in HD-treated healthy ASCs (Figure 4F); however, it did not affect lipedema cells. Additionally, HD media significantly increased the gene expression of all the estrogen-metabolizing enzymes in healthy and lipedema ASCs compared to untreated cells (Figure 4). E2 treatment decreased the expression of LIPE and SULTE1 in HD-treated healthy ASCs but not STS, suggesting that healthy cells respond to estrogen treatment.

### 3.4. Estrogen Significantly Increased the Expression of PPAR-ϒ2 in Differentiated Lipedema Adipocytes and Spheroids

E2 treatment significantly increased the protein expression of PPAR-γ2 in differentiated lipedema adipocytes both in 2D monolayer culture (1.5-fold, Figure 5B, Appendix A) and spheroids (3-fold, Figure 6B, Appendix A) compared to untreated control lipedema (Figure 5 and Figure 6). Additionally, PPAR-γ2 protein expression in E2-treated lipedema spheroid is significantly higher than in healthy spheroids (3-fold, Figure 6B).

In 2D culture, E2 treatment significantly increased ERα and ERβ gene expression in estrogen HD-treated healthy adipocytes (~2-fold, Figure 5B) compared to untreated healthy and lipedema cells. In contrast, E2 treatment significantly increased ERα gene (1.5-fold) and protein (5-fold) expression in treated differentiated lipedema spheroids compared to untreated control cells and healthy spheroids (Figure 6B,D, Appendix A). E2 treatment also increased ERβ protein expression in HD-treated differentiated lipedema spheroids compared to corresponding healthy cells (~6-fold, Figure 6D, Appendix A).

## 4. Discussion

Lipedema manifests during puberty or upon other significant hormonal changes [5,14]. We hypothesized that alteration in sex-specific hormones, in particular estrogen, drives lipedema pathogenesis. Estrogen and its receptors have been shown to play a role in adipose tissue development, physiology, and function [35,36,37,38]. Studies have shown that estrogen is protective against abdominal obesity [39] and bone development [40].

In this study, the effect(s) of estrogen treatment (E2) on the proliferation and stemness of ASCs as well as on the expression of ERs, ER metabolizing enzymes, and adipogenic markers in ASCs and differentiated adipocytes in 2D monolayer and 3D (spheroid) cultures were investigated. The data revealed that E2 increased the proliferation and the expression of stemness markers, CD73, in healthy but not in lipedema ASCs in 2D monolayer culture (Figure 1), suggesting that lipedema ASCs are not sensitive to estrogen treatment. Our data also showed that E2 treatment increased ERα and ERβ expression in healthy and lipedema ASCs in 2D monolayer culture (Figure 2). In 3D culture, E2 treatment decreased the expression of ERα and GPER in lipedema but not in healthy spheroids, indicating that lipedema spheroids are responsive to the E2 treatment (Figure 3). Studies have shown that cells grown in 3D culture mimic the in vivo microenvironment [41,42,43], and thus, E2 treatment might be experienced differently by cells grown in 3D compared to 2D monolayer cultures due to cell–cell and the cell–ECM interactions. In addition to the different culture conditions, the variability between the cell lines is considered one of the major factors affecting the experiments conducted in this study.

Furthermore, studies have shown that E2 treatment regulates the expression of adipogenesis-related transcription factors [44], and contributes to adipocyte hyperplasia [13,45]. In our study, estrogen treatment significantly increased the expression of PPAR-ϒ2 in differentiated lipedema adipocytes and spheroids (Figure 5 and Figure 6), which correlates with the increase in adipocyte size observed in lipedema tissue [10].

Hydroxysteroid 17-beta dehydrogenase 7, HSD17B7, is an enzyme that converts estrone (E1, less active form of estrogen) to estradiol (E2, active form). Studies have shown that the HSD17B7 gene is highly expressed in breast cancer cells, causing the progression of breast cancer [46,47,48]. In our study, E2 treatment significantly increased the expression of HSD17B7 in lipedema ASCs but not in healthy cells (Figure 4), suggesting a role of estradiol in lipedema pathogenesis similar to breast cancer. Another key enzyme in the estrogen cycle is hormone-sensitive lipase (LIPE) [35,49,50]. Our data showed that E2 treatment increases LIPE gene expression in HD-lipedema adipocytes and in adipogenic-differentiated spheroids but not in healthy ASCs (Appendix A). Studies have shown that an increase in LIPE is correlated with increased obesity and adipocyte hypertrophy [51,52,53], a phenomenon observed in lipedema [10].

Estrogen sulfatase (STS) and estrogen sulfotransferase (SULTE1) enzymes regulate estrogen synthesis [54,55]. The data showed that E2 treatment increases STS but not SULTE1 gene expression in HD-lipedema ASCs but not healthy ASCs (Figure 4). Studies have demonstrated that in breast carcinoma, there is an increased expression of STS along with an elevated expression of the HSD17B enzymes. This increased expression leads to the increased synthesis of estradiol, a potent form of estrogen. Consequently, the inhibition of STS is now being considered as a potential therapeutic strategy for hormone-dependent diseases [56,57,58]. Furthermore, estrogen treatment increased the expression of aromatase gene CYP19A1, an enzyme that converts androgens to estradiol, in healthy ASCs but not in lipedema. Studies have shown that alteration in CYP19A1 expression is associated with adipose tissue inflammation and the development of metastasis in breast cancer [59,60,61].

Taken together, these findings suggest that estrogen plays a role in the pathogenesis of lipedema in a similar manner to estrogen-related diseases, such as breast cancer. Additionally, the change we observed is statistically significant based on our current sample size. The number of samples used in our study is relatively low, which means that there is a higher chance that our results could be influenced by random variation or other factors. To ensure the reliability and confirm the biological significance of these findings, it is crucial to collect a larger and more representative sample size. Gathering additional samples will strengthen our analysis by reducing the margin of error and increasing the robustness of our conclusions based on the data.

This study has two main limitations: (1) the small number of samples due to the limited availability of ASCs from lipedema participants, and (2) the lipedema samples used in this study were clustered in Stage 2. Therefore, our future studies will include a larger cohort encompassing all stages of lipedema and menopausal status to comprehensively investigate the role of estrogen in subcutaneous adipose tissue. This investigation aims to determine the expression and activity of estrogen in pre- and post-menopausal women with lipedema, thereby understanding the pathophysiology of lipedema and potentially characterizing it as a hormonal disease.

## 5. Conclusions

Lipedema is a complex disease; in addition to hormonal components, inflammation, fibrosis, and adipose tissue angiogenesis contribute to the disease’s severity. This study indicates that the expression of ERs and several estrogen-metabolizing enzymes are different in lipedema and suggests that estrogen may play a role in adipose tissue dysregulation in lipedema. Exploring this possible etiology further could contribute to expanding treatment options and management available for lipedema. Thus, developing a potential treatment for lipedema should take into consideration inhibitors of ER-metabolizing enzymes, inflammation, and fibrosis.

## Figures and Tables

**Figure 1 biomedicines-12-01042-f001:**
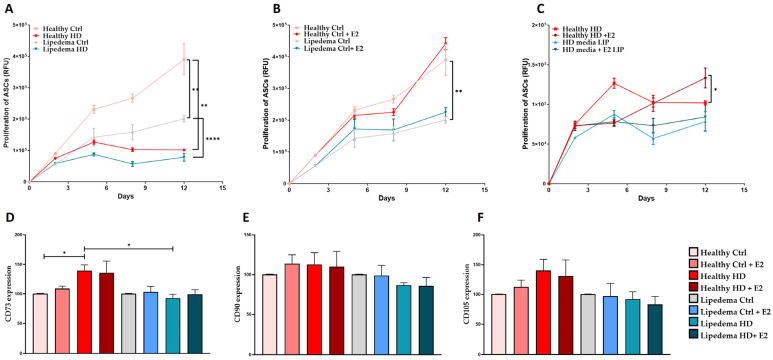
Effect of 17β-Estradiol on the proliferation and stemness of ASCs in 2D monolayer culture. (**A**–**C**). The AlamarBlue assays show a significant decrease in the proliferation rate of both healthy and lipedema ASCs treated with HD media (**A**), no change in cells seeded in DMEM-F12 and treated with E2 (**B**), and an increase in the proliferation rate of HD healthy ASCs treated with E2 compared to lipedema cells (**C**) at day 12 in culture (*n* = 3). RFU: Relative fluorescence unit. The values are the mean SEM. ** *p* < 0.01, **** *p* < 0.0001. (**D**–**F**). Flow cytometry analysis of the surface marker expression shows a significant increase in (**D**) CD73 in HD healthy ASCs compared to untreated healthy and lipedema ASCs (*n* = 3). Values are means ± SEM. * *p* < 0.05.

**Figure 2 biomedicines-12-01042-f002:**
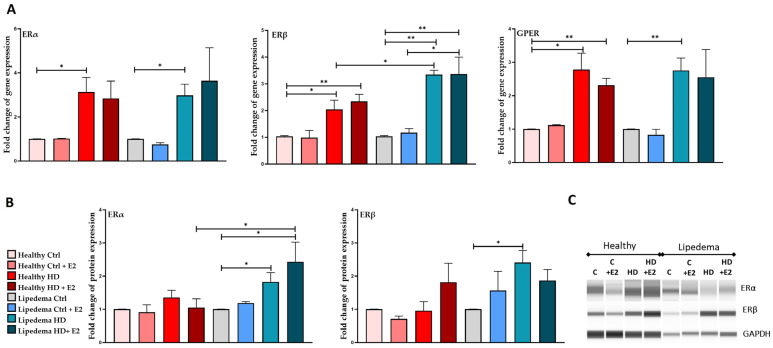
Expression of estrogen receptors in ASCs. (**A**). qRT-PCR shows a significant increase in ER gene expression in HD-treated healthy and lipedema ASCs (*n* = 3). (**B**). Quantification of Western Blot gels shows increased ER protein expression in HD-treated healthy and lipedema ASCs (*n* = 3). (**C**). Capillary Western blot (Jess) assay showing ERα, ERβ, and GADPH protein expression in an assembled gel-like image view. Values are means ± SEM. * *p* < 0.05; ** *p* < 0.01.

**Figure 3 biomedicines-12-01042-f003:**
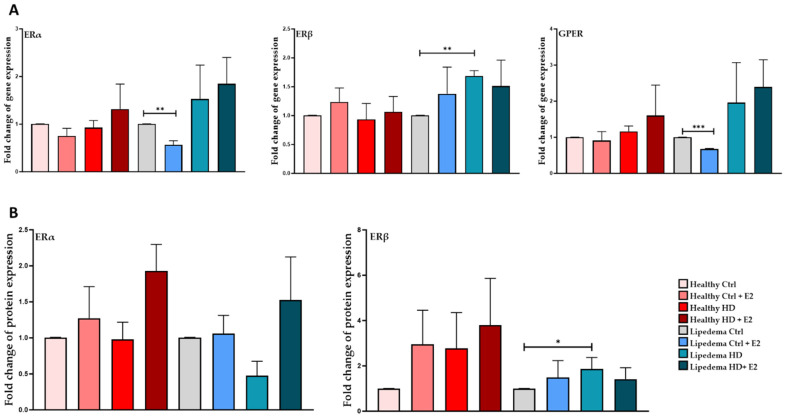
Expression of estrogen receptors in spheroids. (**A**). qRT-PCR shows a significant decrease in ERα and GPER and an increase in ERβ gene expression in HD-treated lipedema spheroids (*n* = 3). (**B**). Quantification of Western Blot gels shows increased ERβ protein expression in HD-treated lipedema spheroids (*n* = 3). Values are means ± SEM. * *p* < 0.05; ** *p* < 0.01; *** *p* < 0.001.

**Figure 4 biomedicines-12-01042-f004:**
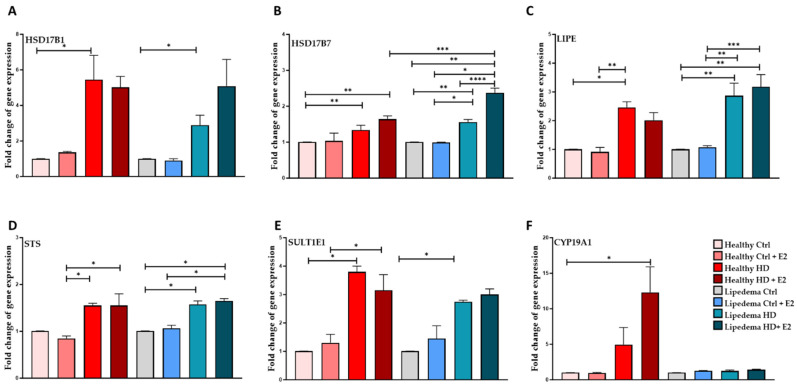
Expression of estrogen-metabolizing enzymes in ASCs. (**A**–**F**). qRT-PCR shows a significant increase in HSD17B1 (**A**), HSD17B7 (**B**), LIPE (**C**), and SULTE1 (**E**) gene expression in HD-treated healthy and lipedema ASCs (*n* = 3). (**B**). qRT-PCR shows a significant increase in HSD17B7 in estrogen HD-treated healthy and lipedema ASCs (*n* = 3). (**C**,**D**). qRT-PCR shows a significant increase in LIPE (**C**), STS (**D**), and CYP19A1 (**F**) gene expression in estrogen HD-treated lipedema ASCs (*n* = 3). (**F**). qRT-PCR shows a significant increase in gene expression in estrogen HD-treated healthy ASCs (*n* = 3). The values are means ± SEM. * *p* < 0.05; ** *p* < 0.01; *** *p* < 0.001; **** *p* < 0.0001. Abbreviations: HSD17B, hydroxysteroid 17-beta dehydrogenase; STS, steroid sulfatase; SULT1E1, estrogen sulfotransferase.

**Figure 5 biomedicines-12-01042-f005:**
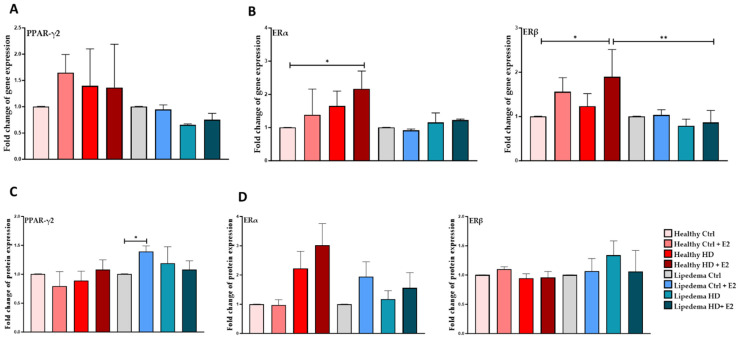
Expression of adipogenic marker and ERs in differentiated healthy and lipedema adipocytes. (**A**). qRT-PCR shows the expression of PPAR-γ2 in differentiated cells (*n* = 3). (**B**). qRT-PCR shows a significant increase in ER gene expression in estrogen HD-treated cells in healthy adipocytes (*n* = 3) and a significant decrease in ERβ gene expression in estrogen HD-treated cells lipedema cells compared to healthy cells. (**C**). Quantification of Western Blot gels shows a significant increase of PPAR-γ2 in estrogen-differentiated lipedema adipocytes. (**D**). Quantification of Western Blot gels of ER in estrogen-differentiated healthy and lipedema adipocytes. Abbreviation: PPAR-γ, peroxisome proliferator-activated receptor gamma. Values are means ± SEM. * *p* < 0.05; ** *p* < 0.01.

**Figure 6 biomedicines-12-01042-f006:**
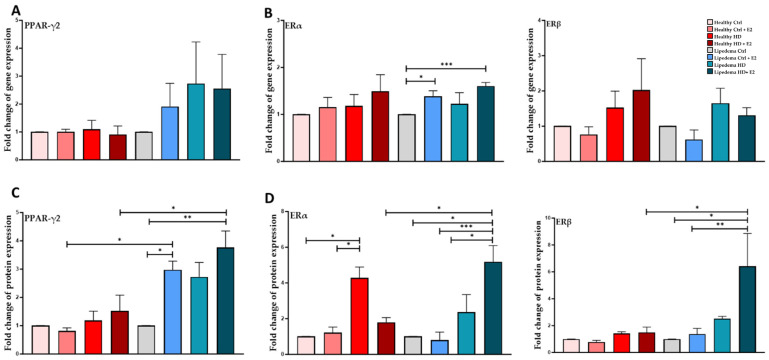
Expression of adipogenic marker and ERs in differentiated healthy and lipedema differentiated spheroids. (**A**). qRT-PCR shows the expression of PPAR-γ2 in differentiated spheroids (*n* = 3). (**B**). qRT-PCR shows a significant increase in ERα gene expression in estrogen HD-treated cells in lipedema-differentiated spheroids (*n* = 3). (**C**,**D**). Quantification of Western Blot gels shows a significant increase of PPAR-γ2 in estrogen-treated differentiated lipedema spheroids. Abbreviation: PPAR-γ, peroxisome proliferator-activated receptor gamma. Values are means ± SEM. * *p* < 0.05; ** *p* < 0.01; *** *p* < 0.001.

**Table 1 biomedicines-12-01042-t001:** Characteristics of healthy and lipedema patients.

Characteristics	Healthy	Lipedema
*n*	5	5
Sex	Female	Female
Age	45 ± 6.42	49.8 ± 3.97
BMI	29.2 ± 2.77	29.9 ± 3.34
Stage 2	−	100%

**Table 2 biomedicines-12-01042-t002:** List of primers used for qRT-PCR.

Name	Forward (5′-3′)	Reverse (5′-3′)
ERα	GCCATGGTGGAGATCTTCGA	CCTCTCCCTGCAGATTCATCA
Erβ	AGAGCTCCTGGTGTGAAGCAA	GACAGCGCAGTGAGCATC
GPER1	TTCCGCGAGAAGATGACCATCC	TAGTACCGCTCGTGCAGGTTGA
PPARG2	AGGCGAGGGCGATCTTG	CCCATCATTAAGGAATTCATGTCATA
GAPDH	CGCTGAGTACGTCGTGGAGTC	GCAGGAGGCATTGCAGATGA
STS	GGACTGGAGTGTGGGGCAGAT	GTGCTCCCTGGTCCGATGTG
LIPE	AGACTTCCGCCTGGGTGCCT	CGGCGCATCGGCTCTGCTAT
CYP19A1	ACTACAACCGGGTATATGGAGAA	TCGAGAGCTGTAATGATTGTGC
HSD17B1	GAGCGTGGGAGGATTGATGG	AGGCTCAAGTGGACCCCAAA
HSD17B7	TTGACACCATATAATGGAACAGAAG	TGATCAGAGGATTGAGAGATTCAG
HSD17B12	GCCAACTTTGGATAAGCCCTCTC	AGGCAGGTTTGAGATTATCGAGC
SULT1E1	TGCCACCTGAACTTCTTCCTGC	CCAGGATTTGGATGACCAGCCA

## Data Availability

Data are contained within the article and Appendix A.

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
