# Peer review of "The Expression of Adipogenic Marker Is Significantly Increased in Estrogen-Treated Lipedema Adipocytes Differentiated from Adipose Stem Cells In Vitro"

_biomedicines, 2024, doi:10.3390/biomedicines12051042_

Round 1

Reviewer 1 Report

Comments and Suggestions for Authors

In this study, we analyzed the gene expression differences of the molecular markers related to estrogen signaling in adipose stem cells (ASCs) and adipocytes of lipoedema patients and healthy individuals, providing a new perspective for future research on the pathogenesis and potential treatment of lipoedema. But there are many things in this paper that need to be modified and perfected. If the preamble is duly compressed; The manufacturers and batch numbers of the main reagents and drugs used for research should be provided; Generally, two levels of significance test are sufficient, and four levels are not necessary. The reasonable inference in the conclusion part should be supported by experiment; It is also necessary to discuss the factors that may affect the experimental results. References should be based on the last three years. Based on the above problems, it is suggested to revise and improve the trial.

Author Response

We thank the reviewer for the suggestion. The lot number of 17β-estradiol E2 and charcoal-stripped (CS) FBS is added to the methods, edited the conclusion, and updated the references.

We have added this phrase to the discussion: The high variability observed between the cell lines and the differences in culture methods are considered the major factors affecting the experiments conducted in this study; both of these factors are now included in the paper. We have also discussed these factors in the discussion section as limitations, and they will be taken into account in designing future trials to improve the reliability of our findings. 

Reviewer 2 Report

Comments and Suggestions for Authors

This paper detected the expression of ER and ER metabolizing enzymes in healthy and Lipedema ASCs. Then the impact of E2 on the proliferation and adipogenic differentiation of ASC was assayed. The work is helpful to understand the role of E2 in adipose tissue dysregulation in lipedema although the data are not enough to resolve it.

Several questions,

1) The author detected many changes in gene expression, some of which were more prominent, while others, despite being statistically significant, were not large enough, especially for qPCR data. As this method is relatively sensitive, changes that are not significant enough (such as less than a twofold change) may not have significant biological meaning. It is hoped that the author will fully consider this point when summarizing the conclusions.

2) Concerning the description of the qPCR method, the author did not indicate how to avoid potential genomic DNA contamination in cDNA. Additional information is desired.

3) The statements 'Institutional Review Board Statement: Not applicable.' and 'Informed Consent Statement: Not applicable.' are unacceptable.

Author Response

We thank the reviewer for their astute comments and suggestions, which we have addressed in the revised manuscript. All concerns have been discussed. In the following sections, we have addressed the comments specific to each reviewer.

Response to Reviewer:

1) The author detected many changes in gene expression, some of which were more prominent, while others, despite being statistically significant, were not large enough, especially for qPCR data. As this method is relatively sensitive, changes that are not significant enough (such as less than a twofold change) may not have significant biological meaning. It is hoped that the author will fully consider this point when summarizing the conclusions.

Response:

We thank the reviewer for highlighting this point. We have added this comment to the discussion:

“Additionally, the change we observed is statistically significant based on our current sample size; the number of samples used in our study is relatively low. This means there is a higher chance that our results could be influenced by random variation or other factors. Collecting a larger and more representative sample size is crucial to ensure reliability and confirm the biological significance of these findings. Gathering additional samples will strengthen our analysis by reducing the margin of error and increasing the robustness of our conclusions based on the data”.

2) Concerning the description of the qPCR method, the author did not indicate how to avoid potential genomic DNA contamination in cDNA. Additional information is desired.

Response:

The RNeasy Extraction Kits, from Qiagen, have RNeasy silica-membrane technology that efficiently eliminates the genomic DNA without DNase treatment.

3) The statements’ Institutional Review Board Statement: Not applicable.’ and ‘Informed Consent Statement: Not applicable.’ are unacceptable.

Response:

The cell lines used in this study were characterized in a previous paper we published in 2020 (reference #29):

Al-Ghadban S, Diaz ZT, Singer HJ, Mert KB, Bunnell BA. Increase in Leptin and PPAR-γ Gene Expression in Lipedema Adipocytes Differentiated in vitro from Adipose-Derived Stem Cells. Cells. 2020;9(2):430.

The information needed is mentioned in the Materials and Methods section of the article mentioned above:

All subjects provided their informed consent for inclusion before participating in the study. The study was conducted in accordance with the Declaration of Helsinki, and the Human Research and Protection Program approved the protocols at the University of Arizona (Institutional Review Board (IRB) protocol number: 1602399502) and Tulane University (IRB protocol number: 9189). All patients’ identifiers were kept confidential, and all received samples utilized in this study were de-identified.”

If the reviewer believes mentioning them in this paper would be beneficial, we should do so.